# Intestinal Microbiota and Its Effect on Vaccine-Induced Immune Amplification and Tolerance

**DOI:** 10.3390/vaccines12080868

**Published:** 2024-08-01

**Authors:** Yixin Liu, Jianfeng Zhou, Yushang Yang, Xiangzheng Chen, Longqi Chen, Yangping Wu

**Affiliations:** 1Department of Pulmonary and Critical Care Medicine, West China Hospital, Sichuan University, Chengdu 610041, China; drliuyx@stu.scu.edu.cn; 2Department of Thoracic Surgery, West China Hospital, Sichuan University, Chengdu 610041, China; 2020224020107@stu.scu.edu.cn (J.Z.); drchenlq@scu.edu.cn (L.C.); 3Department of Liver Surgery & Liver Transplantation, West China Hospital, Sichuan University, Chengdu 610041, China; xiangzheng.chen@wchscu.cn; 4State Key Laboratory of Respiratory Health and Multimorbidity, West China Hospital, Chengdu 610041, China

**Keywords:** intestinal microbiota, vaccine, immune amplification, immune tolerance, vaccine design, outer membrane vesicles

## Abstract

This review provides the potential of intestinal microbiota in vaccine design and application, exploring the current insights into the interplay between the intestinal microbiota and the immune system, with a focus on its intermediary function in vaccine efficacy. It summarizes families and genera of bacteria that are part of the intestinal microbiota that may enhance or diminish vaccine efficacy and discusses the foundational principles of vaccine sequence design and the application of gut microbial characteristics in vaccine development. Future research should further investigate the use of multi-omics technologies to elucidate the interactive mechanisms between intestinal microbiota and vaccine-induced immune responses, aiming to optimize and improve vaccine design.

## 1. Introduction

Vaccination plays a pivotal role in public health, significantly reducing the incidence rates of diseases such as measles, mumps, and rubella by 90% to 100% since the 1960s [1]. Furthermore, vaccination has shown potential in cancer prevention, with the human papillomavirus (HPV) vaccine and the hepatitis B virus (HBV) vaccine effectively preventing cancers related to chronic infections [2]. However, not all individuals respond uniformly to vaccines, presenting a significant challenge in preventive medicine and global health.

The heterogeneity in vaccine responses is influenced by various host factors, including age, gender, genetic background, and overall health, which can significantly impact the immune response to vaccines. For instance, elderly individuals often exhibit weaker responses to vaccines due to immunosenescence [3]. Genetic factors may also contribute to inter-individual variability in vaccine responsiveness, leading to reduced or absent immunogenicity to certain vaccines [4]. The most generally recorded non-response has been that of the HBV vaccine, with up to 10% of vaccinated individuals failing to develop the protective antibody level against the HBV surface antigen present in the vaccine [5].

The functional status of the immune system influences both antibody production and cell-mediated immune responses following vaccination. Interestingly, the correlation between antibody levels and protective efficacy is not always straightforward. Individuals may achieve effective immunity through other immunological mechanisms despite having lower antibody levels [6]. Immunological memory function is crucial in vaccination as it provides long-term protection through the rapid response of memory cells.

Emerging research highlights the significant role of intestinal microbiota in modulating immune responses and influencing vaccine efficacy [7]. As the largest microbial community in the human body, intestinal microbiota interacts with immune cells and can enhance antibody responses following vaccination through pattern recognition receptors (PRRs) [8]. The administration of antibiotics may disrupt the intestinal microbiota, consequently attenuating the immune response to vaccines [6].

The impact of intestinal microbiota is particularly notable in the development and application of COVID-19 vaccines. In a healthy intestinal microbiota environment, the immunogenicity of the COVID-19 vaccine can be enhanced through various mechanisms (Figure 1), substantially decreasing the severity and mortality rates in COVID-19 patients, especially within the elderly demographic [9,10,11]. Moreover, differences in microbial environments across geographical regions may explain variations in COVID-19 vaccine responses and efficacy [7]. Hence, the role of intestinal microbiota in vaccine design and efficacy represents a promising avenue for improving immunogenicity and personalizing medical approaches. Elucidating the interplay between the intestinal microbiota and vaccine-induced immune responses holds critical importance in refining vaccination protocols and augmenting vaccine efficacy.

This review provides current insights into the interplay between the intestinal microbiota and the immune system, focusing particularly on its intermediary function in vaccine efficacy. Additionally, it summarizes strategies for leveraging intestinal microbiota in designing or delivering vaccines to enhance their effectiveness.

## 2. The Influence of Intestinal Microbiota on the Immune System and Host Immune Responses

### 2.1. Intestinal Microbiota and Its Potential Relationship with Immune Responses

The intestinal microbiota refers to the community of bacteria residing in the human intestine. These microorganisms play a crucial role in maintaining host health by participating in various physiological processes, including digestion, metabolism, and immune regulation. Numerous studies have explored the complex relationship between intestinal microbiota and immune responses. The intestinal microbiota influences the host’s immune responses through multiple mechanisms, including but not limited to the modulation of immune cell activity, enhancement of antigen presentation, and regulation of inflammatory responses [12].

The intestinal microbiota can promote the maturation and optimal functioning of the immune system through interactions with intestinal epithelial cells and immune cells. Certain probiotics can enhance intestinal barrier function by interacting with epithelial cells, thereby preventing pathogen invasion. Zhou et al. highlighted the role of intestinal microbiota in shaping the mucosal immune system within the intestinal region [12]. The interaction between the intestinal microbiota and the immune system is bidirectional; the microbiota can affect the proliferation, differentiation, and function of immune cells, while diseases can alter the composition of the microbiota [12]. Recent findings further emphasize the role of intestinal microbiota in immune regulation. Ivo et al. demonstrated that intestinal microbiota regulates the epigenetic programming of innate lymphoid cells (ILCs), influencing their differentiation and function [13]. This highlights the microbiota’s fundamental ability to affect immune cell development and responses. The relationship between intestinal microbiota and immune responses could extend to a variety of diseases. Zheng et al. studied respiratory syncytial virus (RSV) infection and found that RSV-induced changes in intestinal microbiota disrupt the balance between Th1/Th2 and Treg/Th17 cells, exacerbating immune responses and clinical symptoms [14]. This underscores the importance of maintaining a stable intestinal microbiota to prevent immune dysregulation during infections. Intestinal microbiota also plays a critical role in autoimmune diseases. Studies by De et al. [15] and Selma et al. [16] have shown that alterations in intestinal microbiota are associated with diseases such as systemic lupus erythematosus and rheumatoid arthritis, respectively. These changes can influence immune responses, leading to disease onset and progression. Additionally, the impact of intestinal microbiota on immune responses is evident in cancer. Ritis et al. discussed the secretion of cytokines IL-6 and IL-8 associated with intestinal inflammatory diseases caused by sleep deprivation may be the cause of prostate cancer development by modulating immune responses and promoting a tumor-friendly environment [17]. 

Furthermore, the intestinal microbiota can regulate the activity and function of immune cells through the production of metabolites such as short-chain fatty acids (SCFAs). Johnson et al. demonstrated that microbial metabolites like SCFAs can modulate immune responses, affecting both local and systemic immunity [18]. These metabolites play a role in maintaining gut-–lung homeostasis and influencing responses to respiratory infections [19]. 

The intestinal microbiota plays a crucial role in antigen presentation. Dendritic cells (DCs) in the gut can capture and process antigens from the intestinal microbiota and present these antigens to T cells, thereby initiating specific immune responses. Studies have shown that the composition and diversity of the intestinal microbiota can significantly affect the function of DCs, thus influencing the host’s immune responses [20]. Additionally, the intestinal microbiota influences the host’s immune status by regulating inflammatory responses. Certain intestinal microbiota can reduce the host’s inflammatory response by inhibiting the production of inflammatory mediators, thereby maintaining immune balance. The lactic acid-producing probiotic *Saccharomyces cerevisiae* regulates macrophage polarization states and inhibits the expression of pro-inflammatory cytokines both in vivo and in vitro. By increasing the activity of monocarboxylate transporter-mediated lactate uptake in macrophages, it suppresses the excessive activation of the NLRP3 inflammasome and downstream caspase-1 pathway, thereby modulating the gut microbiota and alleviating ulcerative colitis [21]. Conversely, dysbiosis can lead to excessive inflammatory responses, triggering various immune-related diseases [20]. In patients with inflammatory bowel disease (IBD), there is a decrease in potentially anti-inflammatory microbes such as *Bacteroidetes* and *Faecalibacterium prausnitzii*, while potentially pro-inflammatory microbes such as Proteobacteria and *Ruminococcus gnavus* increase [22].

In summary, the intestinal microbiota plays a critical role in regulating the host immune system, influencing both local and systemic immunity. By modulating immune cell activity, enhancing antigen presentation, and regulating inflammatory responses, the intestinal microbiota not only affects the host’s immune responses but may also play an important role in the amplification and tolerance induced by vaccines.

### 2.2. Intestinal Microbiota Mediate Immune Tolerance and Diseases

The role of the intestinal microbiota in the host immune system has garnered increasing attention. The intestinal microbiota not only plays a crucial role in maintaining normal immune responses for gut health but also influences immune tolerance through various mechanisms [20].

The intestinal microbiota can regulate the intensity and nature of immune responses through interactions with host immune cells. For example, certain gut bacteria can influence DC function through their metabolites, such as SCFAs, thereby modulating T cell activation and differentiation [20]. This regulation may influence the body’s immune response, leading to tolerance. Kim explored the role of SCFAs in immune tolerance, noting that these metabolites, produced primarily through microbial fermentation of dietary fiber, have tissue- and disease-specific impacts on autoimmune diseases such as type 1 diabetes and multiple sclerosis [23]. These metabolites regulate lymphocyte development and tissue barrier function, crucial for maintaining immune homeostasis and preventing autoimmunity. This aligns with Shao et al.’s emphasis on the dynamic interactions between the intestinal microbiota and the host immune system, particularly the role of microbial metabolites in influencing immune function and homeostasis [24]. Lyu et al. further elucidated this mechanism, showing how group 3 innate lymphoid cells (ILC3) interact with regulatory T cells to establish immune tolerance in the gut. These interactions are essential for selecting microbiota-specific regulatory T cells, preventing inflammatory responses, and maintaining gut health [25]. Bashir et al. studied age-associated gut dysbiosis and its impact on DC tolerance, demonstrating that aging disrupts gut microbial composition, leading to immune dysregulation and a loss of DC tolerance [26]. This finding parallels Kim and Shao et al.’s observations on the importance of microbial balance in maintaining immune homeostasis.

Dysbiosis can lead to abnormal immune tolerance, triggering various diseases. Research has found that the composition and diversity of the intestinal microbiota are closely related to immune-related diseases such as inflammatory bowel disease, autoimmune diseases, and allergic diseases. These diseases may be associated with dysbiosis-induced immune tolerance mechanisms. Akagbosu et al. identified a novel antigen-presenting cell type, Thetis cells, crucial for generating peripheral regulatory T (pTreg) cells in the gut. These cells maintain immune tolerance to the intestinal microbiota, preventing colitis [27]. This finding complements Sanidad et al.’s research, which showed that gut bacteria-derived serotonin plays a significant role in promoting regulatory T cell differentiation, essential for maintaining immune tolerance to dietary antigens and commensal bacteria [28]. Gavzy et al. discussed the mechanism by which *bifidobacteria* regulate immune responses, emphasizing their role in upregulating regulatory T cells and maintaining gut barrier function. This mechanism is crucial for preventing autoimmune reactions and maintaining immune homeostasis [29]. Hou et al. explored the role of intestinal microbiota in Graves’ disease (GD) and Graves’ ophthalmopathy (GO), showing that dysbiosis exacerbates these conditions. They observed significant differences in microbiota composition between GD/GO patients and healthy controls, with increased levels of Lactobacillus and Prevotella in GD patients [30]. In addition, De Filippis et al. identified specific intestinal microbiota characteristics associated with allergies and immune tolerance acquisition in children, observing higher levels of inflammatory microbes in allergic children, suggesting that dysbiosis may hinder the formation of immune tolerance, leading to chronic allergic conditions [31]. This finding echoes Méndez et al.’s focus on the intestinal microbiota’s contribution to immune tolerance in food-allergic infants, noting that modulating the intestinal microbiota with prebiotics and probiotics can induce tolerance, though current evidence remains uncertain [32]. Cheng et al. [33] investigated the impact of maternal *Bifidobacterium bifidum* during pregnancy on offspring intestinal microbiota and immune system development, showing that this treatment can induce immune tolerance to allergens in offspring, which is consistent with Méndez et al.’s [32] proposal of probiotics’ potential in regulating immune responses. 

The manipulation of intestinal microbiota-mediated immune tolerance for the treatment of clinical diseases has been increasingly reported in the literature. Liu et al. reviewed the application of fecal microbiota transplantation (FMT) in managing autoimmune diseases, emphasizing FMT’s potential to restore gut microbial balance and immune tolerance [34], highlighting the importance of microbial-targeted interventions. Köhler and Zeiser [35] discussed the intestinal microbiota’s role in influencing immune tolerance after allogeneic hematopoietic stem cell transplantation and its role in preventing graft-versus-host disease (GVHD). This study underscores the intestinal microbiota’s crucial role in mediating immune tolerance in clinical settings, supporting Liu et al.’s [34] findings on the potential of microbial-targeted interventions.

The interactions between the intestinal microbiota and immune tolerance are intricate, playing significant roles in mediating immune tolerance and causing diseases. In-depth research into the interaction mechanisms between the intestinal microbiota and the host immune system can aid in developing new vaccine strategies and therapeutic approaches to improve the prevention and treatment of immune-related diseases.

## 3. Intestinal Microbiota and Vaccines

### 3.1. Oral Vaccines and Intestinal Microbiota

Oral vaccines are absorbed through the gastrointestinal tract and interact directly with the intestinal microbiota, influencing vaccine-induced immune amplification and tolerance. The intestinal microbiota plays a critical role in maintaining intestinal immune homeostasis, and its diversity and composition can significantly impact the immune system’s response to vaccines.

The intestinal microbiota modulates immune responses through interactions with immune cells. Studies have shown that individuals with a higher abundance of *Clostridiales* and a lower abundance of *Enterobacterales* in their gut had stronger memory B cell responses to the cholera vaccine (OCV) [36]. In addition, the composition and diversity of the intestinal microbiota are crucial for the effectiveness of oral vaccines. A healthy and diverse intestinal microbiota can provide broader immune stimulation, enhancing the immunogenicity of vaccines. Conversely, dysbiosis (such as that caused by antibiotic use) may weaken vaccine efficacy and increase immune tolerance [20]. Magwira and Taylor [37] and Bhattacharjee and Hand [38] discussed the role of the intestinal microbiota in the effectiveness of oral rotavirus (ORV) vaccines. They noted that vaccine efficacy is lower in low-income countries compared to high-income countries, likely due to differences in intestinal microbiota composition. Bhattacharjee and Hand [38] further emphasized that malnutrition and chronic gastrointestinal infections lead to dysbiosis, affecting vaccine efficacy. Both studies concluded that the diversity and health of the intestinal microbiota are critical for vaccine immune responses. Similarly, Yuki et al. conducted a randomized controlled trial verifying the safety and immunogenicity of the oral MucoRice-CTB vaccine in humans, finding a close correlation between intestinal microbiota diversity and vaccine response [39]. In detail, Zimmermann and Curtis reviewed the impact of intestinal microbiota on vaccine responses, noting that higher relative abundances of *Actinobacteria* and *Firmicutes* were associated with better immune responses, whereas *Proteobacteria* and *Bacteroidetes* had the opposite effect [40].

Additionally, the intestinal microbiota can indirectly regulate vaccine absorption and distribution by influencing intestinal barrier function. Certain probiotics can enhance intestinal barrier function, reducing pathogen invasion and thereby increasing vaccine protection. The metabolites of the intestinal microbiota can also affect the tight junctions of intestinal epithelial cells, influencing vaccine absorption efficiency [20]. 

In addition to the intestinal microbiota influencing the response to oral vaccines, vaccines can also affect the composition of the intestinal microbiota. Medeiros et al. studied the impact of oral polio vaccine (OPV) revaccination on the gut and upper respiratory microbiomes of infants, finding significant improvements in intestinal microbiota health post-vaccination [41]. This aligns with Cobb et al.’s research on the impact of an oral *Salmonella*-based vaccine on the intestinal microbiota of NOD mice, which found significant changes in metabolic pathways related to inflammation and immune tolerance post-vaccination. These findings suggest that oral vaccines not only prevent specific diseases but also promote overall health by modulating the intestinal microbiota [42]. Hosomi and Kunisawa reviewed the influence of the gut environment on vaccine immune responses, highlighting the important interactions between intestinal microbiota, diet, and the effectiveness of both systemic and oral vaccines. They proposed that optimizing the gut environment could be a potential strategy to enhance vaccine efficacy [43].

These studies indicate that the composition and diversity of the intestinal microbiota play a key role in the immune responses to various oral vaccines. Future research should continue to explore these mechanisms, identify microbial markers of successful vaccination, and develop targeted interventions to improve vaccine efficacy.

### 3.2. Parenteral Vaccines and Intestinal Microbiota

Recent research has shown that intestinal microbiota influences not only local intestinal immune responses but also systemic immune responses through complex immune networks. This role is particularly significant in the context of parenteral vaccine administration.

The intestinal microbiota regulates immune cells and the immune system by producing immunomodulatory molecules, thereby affecting immune responses in distant tissues. For instance, Arnold et al. demonstrated how a single gut microorganism, such as *Helicobacter hepaticus*, significantly impacted the efficacy of a tuberculosis subunit vaccine. They found that *Helicobacter hepaticus* infection suppressed vaccine-induced immune responses by increasing IL-10 expression, and treatment with anti-IL-10 receptor antibodies restored vaccine efficacy [44]. This finding provides direct evidence of the specific role of intestinal microbiota in immune suppression mechanisms. Jeyanathan et al. studied the impact of parenteral BCG vaccination on lung-resident memory macrophages and gut–lung axis-trained immunity. They found that BCG vaccination induced lung memory macrophages and trained immunity by altering the intestinal microbiota. This indicates that systemic vaccination can induce local immune memory at distant mucosal sites through microbiota-mediated pathways, revealing the potential role of the gut–lung axis in vaccine immunity [45]. These studies validate the general influence of intestinal microbiota on vaccine immunity and provide specific evidence of different mechanisms. Similarly, Ng et al. indicated that intestinal microbiota might also influence the immune response to COVID-19 vaccines by modulating angiotensin-converting enzyme 2 (ACE2) expression and altering the secretion of immunoregulatory molecules such as tryptophan, SCFAs, and secondary bile acids, suggesting common microbiota-mediated mechanisms across different vaccines [46].

The intestinal microbiota can indirectly influence systemic immune responses by regulating intestinal barrier function. A healthy intestinal microbiota maintains the integrity of the intestinal barrier, preventing pathogens and toxins from entering the bloodstream and thereby reducing systemic inflammatory responses. After vaccination, maintaining the integrity of the intestinal barrier is crucial for appropriate immune responses, as excessive inflammation can lead to immune tolerance. Ray et al. investigated the impact of intestinal microbiota on immune responses to COVID-19 mRNA vaccines, particularly in healthy individuals and those with HIV. They found that individuals with high IgG titers had lower intestinal microbiota diversity, with specific bacteria such as *Bifidobacterium* and *Enterococcus faecalis* associated with better immune responses, further emphasizing the role of specific bacteria in enhancing vaccine immune responses. These results suggest that modulating the microbiota can significantly improve immune function, especially in immunocompromised populations [47].

Moreover, the intestinal microbiota can directly influence immune responses through interactions with gut-associated lymphoid tissue (GALT). GALT is a crucial component of the intestinal immune system, responsible for monitoring and responding to antigens in the gut. The intestinal microbiota can affect vaccine-induced immune responses by modulating the activity of immune cells within GALT. For example, certain probiotics can enhance the production of IgA antibodies within GALT, thereby improving parenteral vaccine efficacy [48].

The intestinal microbiota plays a critical role in shaping immune responses to parenteral vaccines. By regulating immune cells and molecules, intestinal barrier function, and immune responses within GALT, further research into this relationship could lead to microbiota-based interventions that significantly enhance vaccine efficacy, contributing to better global health outcomes.

## 4. Intestinal Microbiota to Vaccine Effect: Immune Amplification or Tolerance

The intestinal microbiota modulates the host’s immune response through several mechanisms, mainly including the synthesis of metabolic compounds, promotion of intestinal epithelial barrier integrity, regulation of immune homeostasis, and prevention of pathogen colonization [49,50]. Leung et al. found that a deficiency in gut bacteria like *Faecalibacterium prausnitzii* and *Eubacterium rectale* was linked to COVID-19 severity, and their supplementation might hold therapeutic promise [51]. Disruptions in the microbiota can cause irregular immune responses, leading to multiple immune-related disorders [52]. Furthermore, intestinal microbiota metabolites, including short-chain fatty acids and tryptophan metabolites, can directly or indirectly impact immune cell function and distribution, thus regulating the strength and nature of immune responses [53].

In terms of vaccine immunogenicity, the composition and functionality of the intestinal microbiota are recognized as important factors affecting vaccine efficacy. Some gut microorganisms can boost vaccine immunogenicity and protective effectiveness by regulating the host’s immune response [54]. Nutritional status plays a vital role in shaping the intestinal microbiota and the efficacy of vaccines. A balanced diet rich in fiber contributes to a healthy intestinal microbiome, while malnutrition, especially a lack of protein, can cause dysbiosis and affect the immune response to vaccines [55]. Variations in the intestinal microbiota are closely linked to differences in vaccine efficacy. Vaccination following antibiotic treatment may lower antigen-specific IgG levels, with this effect being more noticeable in infants and young children [56,57,58]. Additionally, the intestinal microbiota can serve as a natural adjuvant, inducing cross-reactivity by carrying epitopes similar to vaccine antigens, thereby affecting vaccine immunogenicity [59,60,61]. Intestinal microbiota dysbiosis may also cause a reduction in vaccine immune responses, which can be reversed with fecal microbiota transplantation [57]. The intestinal microbiota plays a significant role in modulating the host immune environment and influencing the effectiveness of vaccines. Understanding these mechanisms is essential for optimizing vaccine efficacy and developing new immunization strategies.

### 4.1. Amplification of Vaccine Efficacy by Intestinal Microbiota

The intestinal microbiota enhances vaccine immunogenicity through various mechanisms, thereby amplifying vaccine efficacy. Primarily, the intestinal microbiota can influence the immune response to vaccines by modulating the development and function of the immune system. For instance, certain probiotics such as *Bifidobacterium* and *Lactobacillus* can improve vaccine efficacy by regulating the gut environment, promoting antibody production, and enhancing T-cell responses [51,55,62].

In COVID-19 vaccine research, it has been found that the intestinal microbiota composition significantly impacts immune responses to vaccines. High concentrations of *Bifidobacterium* in the gut are related to increased neutralizing antibody generation for the inactivated virus vaccine CoronaVac, while the abundance of other strains such as *Bacteroides* and *Ruminococcus* is connected to lower responsiveness [63]. In addition, the efficacy of BNT-162b in terms of antibody production can be boosted by the richness of *Faecalibacterium* and *Akkermansia* in the intestinal microbiota [63]. The intestinal microbiota also plays a role as a natural adjuvant by interacting with the host immune system. Certain gut microbial strains can efficiently regulate DC function, leading to the direct or indirect activation of antigen-presenting cells [59,60]. Furthermore, TLR5-mediated detection of intestinal microbiota-derived flagellin has also been proven to be necessary for generating the optimal antibody response to adjuvant-free influenza vaccines [64]. Butyrate and other SCFAs, generated through the fermentation of fiber by colonic bacteria, also exhibit anti-tumor effects [46,65,66].

A brief summary of the roles of various microbial species that may enhance vaccine efficacy can be found in Table 1 [36,67,68,69,70,71,72,73,74].

### 4.2. Intestinal Microbiota and Immune Tolerance Response to Vaccine

Intestinal microbiota dysbiosis can diminish vaccine immunogenicity, leading to inadequate immune responses. In animal models, intestinal microbiota dysbiosis induced by antibiotic treatment is significantly linked to lower vaccine efficacy. For example, after vaccination, mice and rhesus macaques treated with vancomycin exhibit substantially reduced serum levels of antigen-specific IgG [75]. Reinstating intestinal microbiota diversity can counteract the vaccine hypo-responsiveness induced by vancomycin, emphasizing the crucial role of intestinal microbiota diversity in preserving regular immune responses [75,76]. Tumes et al. reported that young mice, after exposure to ampicillin and neomycin, displayed marked impairment in antibody responses to five diverse vaccines, yet these responses were restored through fecal microbiota transplantation [77]. In clinical trials, antibiotic-induced dysregulation of the intestinal microbiota has been shown to impact vaccine effectiveness. Adults treated with antibiotics exhibit significantly reduced antibody responses to trivalent inactivated influenza vaccine (TIV), particularly in those with lower pre-existing immunity to influenza viruses [78,79]. The administration of antibiotics is correlated with variations in the immunogenicity of rotavirus vaccines [80]. Moreover, Bremel et al. reported that the presence of numerous TCEMs (T cell exposure motifs) in Burkholderia, which are analogous to human immunoglobulin V regions, can diminish vaccine efficacy [81]. Williams et al. demonstrated that Escherichia coli can elicit the production of non-neutralizing antibodies against the HIV gp41 protein [82].

Intestinal microbiota composition and diversity not only impact vaccine efficacy but may also elucidate differences in vaccine effectiveness observed between geographical regions. Reduced oral vaccine responses in low-income countries may be attributed to variations in intestinal microbiota composition within these areas. Such differences might be influenced by factors such as environmental conditions, socioeconomic status, nutritional intake, and sanitation practices [83]. Disruptions in the intestinal microbiota may also impact the long-term immune memory of vaccines. For instance, dysbiosis of the intestinal microbiota induced by broad-spectrum antibiotic treatment significantly reduces the number of memory T cells in the lungs and secondary lymphoid organs following BCG vaccination in mice and inhibits the secretion of interferon-gamma and tumor necrosis factor-alpha [64,84].

A brief summary of the roles of microbial species that may potentially reduce vaccine efficacy can be found in Table 2 [81,82,85].

## 5. Leveraging the Intestinal Microbiome for Vaccine Design

### 5.1. Fundamental Principles of Vaccine Sequence Design

The principle of vaccine design lies in mimicking specific antigenic components of pathogens to induce an immune response in the host against the pathogen, thus providing protection upon future exposure to the same pathogen. In the design process, selecting appropriate vaccine antigens and adjuvants is crucial as it helps optimize the formation and persistence of immune memory [86]. The application of nanotechnology in vaccine design demonstrates significant advantages, particularly in enhancing immune responses through the presentation of multivalent antigens [87]. Furthermore, the utilization of immunoinformatics tools has further enhanced the efficiency and precision of vaccine design, enabling the development of multi-epitope vaccines [88]. Reverse vaccinology approaches utilize genomic information to predict all antigens of a pathogen, enabling the identification of potential vaccine candidates through computer-based analysis, without the need for culturing the pathogen, thus facilitating vaccine development [89]. Research on the novel tuberculosis vaccine MP3RT demonstrates high concordance between immunoinformatics predictions and animal experimental results, validating the potential application of this technology in reverse vaccinology [90]. Structure-based vaccine design involves the identification and redesign of antigenic epitopes by determining the three-dimensional structure of antigens or antigen-antibody complexes, aiming to enhance vaccine stability and immunogenicity [91,92]. However, vaccine design also needs to consider individual variations to ensure the efficacy and safety of vaccines across diverse populations [93].

### 5.2. Utilizing Intestinal Microbiota Features in the Vaccine Design

The intestinal microbiota has demonstrated significant potential in optimizing vaccine design. Following the administration of inactivated and mRNA vaccines, participants with higher abundances of *P. copri* and Megamonas in their intestinal microbiota experienced fewer adverse events [63]. Research by Collins et al. suggests that the microbiota co-evolves with the host, finely tuning the unique needs of the gut, and these microorganisms may offer highly adapted tissue-specific adjuvants [94]. Kim et al. also discovered that metabolites produced by the intestinal microbiota can fuel the host’s antibody response, thereby enhancing the immune efficacy of vaccines [95].

In vaccine development, the utilization of immunoinformatics tools to select specific antigenic epitopes, coupled with the regulatory effects of the intestinal microbiota, can lead to the development of more effective multi-epitope vaccines. In the design of a multi-epitope vaccine targeting *P. anaerobius*, multiple CTL, HTL, and LBL epitopes were selected using immunoinformatics tools, and a 516-amino acid multi-epitope vaccine was constructed, demonstrating favorable characteristics [88]. Regarding vaccine stability, the genetic sequences of the intestinal microbiota also hold potential application value. Future research could utilize multi-omics techniques to comprehensively analyze the interaction between the intestinal microbiota and vaccine immune responses. Utilizing multi-system vaccinology tools such as gene expression, protein or lipid synthesis, and metabolite changes can help understand the comprehensive network and dynamic changes in vaccine-induced immune responses [96]. Hao et al. found that an increased abundance of *Lachnospiraceae* and *Enterococcaceae* members is associated with hematopoietic recovery and gastrointestinal repair following radiation exposure through multi-omics microbes and metabolites analyses of radiation survivors [97]. Additionally, paired microbiome and metabolome analyses can reveal diet-induced changes in the microbiome and metabolome during disease progression. Ting et al. demonstrated the relationship between *Escherichia coli* and *Ruminococcus* and the amino acid conjugation of the bile acid cholic acid in this multi-omics dataset, highlighting their potential role in promoting colorectal cancer [98]. Multi-level analysis will help to elucidate the specific mechanisms of intestinal microbiota in disease progression and vaccine design.

A brief summary of the roles of potential microbial species that may optimize vaccine design can be found in Table 3.

## 6. Potential of the Vaccine Utilizing Microbiota-Derived Outer Membrane Vesicles

The efficacy of vaccines relies not only on their antigenic components but also heavily on how these antigens are delivered to target cells and tissues. Traditional vaccine carriers such as viral vectors and protein carriers have achieved significant success in vaccine development. However, with technological advancements, new carriers and delivery systems continue to emerge, providing more possibilities for vaccine design and application [105]. The delivery system of mRNA vaccines is particularly crucial as an emerging vaccine modality [106]. Due to the inherent instability of mRNA, an effective delivery system is required to protect its stability and ensure its efficient delivery in vivo. Lipid nanoparticles (LNPs), as an advanced mRNA delivery platform, have been successful in the development of COVID-19 vaccines [107].

Of note, specific components of the intestinal microbiota are considered potential vaccine carriers. The intestinal microbiota can communicate with host cells through the outer membrane vesicles (OMVs) they produce, which can carry antigens and stimulate the host’s immune response [108,109]. Utilizing *Escherichia coli*-derived OMVs as an antigen delivery platform effectively induces protective immune responses against pathogens such as *Pseudomonas aeruginosa* and *Acinetobacter baumannii* [108,110]. This delivery mode not only efficiently elicits humoral and cellular immunity but also enhances the recognition and response of the immune system by carrying various pathogen-associated molecular patterns (PAMPs) [109]. However, this delivery method also faces some challenges. Firstly, although strategies have been developed to load recombinant antigens onto OMVs, their universal applicability still needs further validation. Secondly, optimizing the localization and expression of antigens in OMVs to maximize immune response efficacy remains an area requiring further investigation. Additionally, the production and purification processes of OMVs also need to ensure consistency and efficiency to meet the demands of large-scale applications [110]. Moreover, for bacterial strains for which no vaccines currently exist, employing their OMVs as vaccine candidates presents a promising strategy to manage and reduce their pathogenic burden [111].

Interestingly, OMVs from different microbial populations can also modulate host immune responses. For example, *Salmonella Typhimurium* OMVs can stimulate B cells and T cells to initiate the production of specific antibodies [112]. Conversely, OMVs from *Helicobacter pylori* and *Neisseria gonorrhoeae* can inhibit T lymphocyte proliferation [113,114,115]. Selecting appropriate OMVs from the intestinal microbiota as carriers and utilizing modified OMVs (such as encapsulating OMVs with chitosan for protection or incorporating antigen presentation on the OMV surface) can further enhance vaccine efficacy (Figure 2).

## 7. Summary and Conclusions

The role of intestinal microbiota in vaccine design demonstrates its significant potential in modulating immune responses and enhancing vaccine efficacy. Immunoinformatics tools and outer membrane vesicles used as novel vaccine carriers represent diverse and innovative development pathways. Future research should further explore the complex interactions between microbiota and the host immune system to optimize vaccine design and advance personalized medicine.

## Figures and Tables

**Figure 1 vaccines-12-00868-f001:**
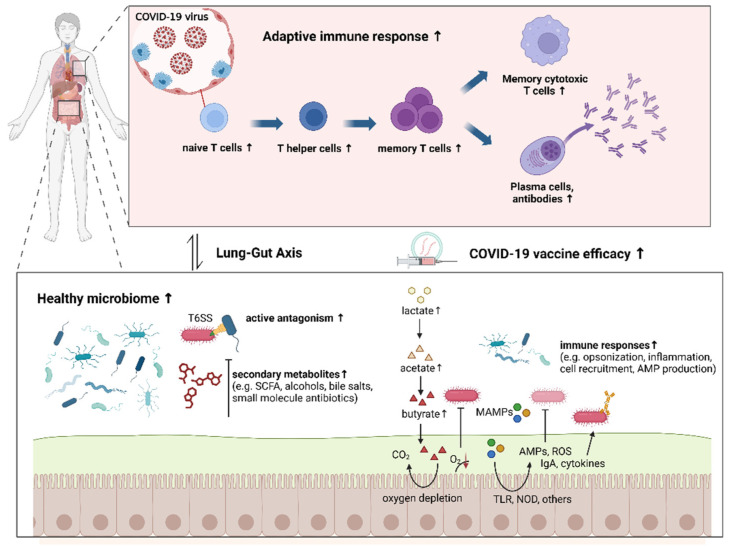
Immunogenicity of the COVID-19 vaccine can be enhanced by healthy intestinal microbiota through various mechanisms.

**Figure 2 vaccines-12-00868-f002:**
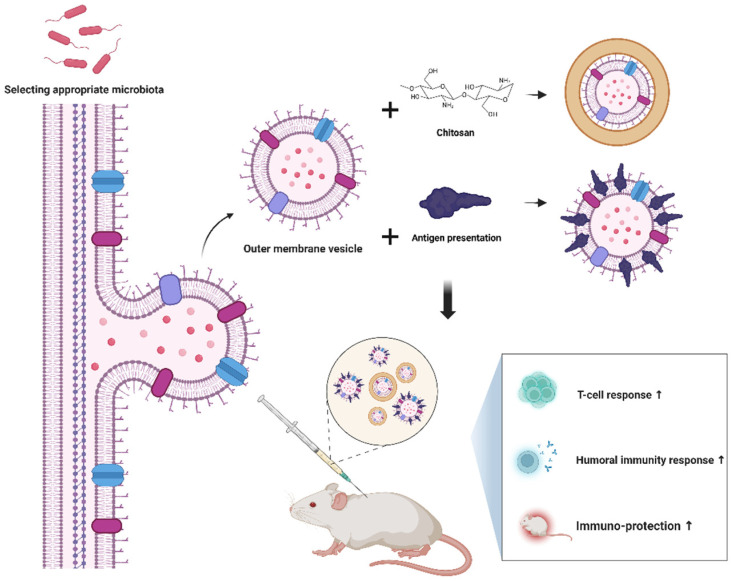
Outer membrane vesicle designs hold the potential to significantly augment the efficacy of vaccines.

**Table 1 vaccines-12-00868-t001:** Microbial species that may enhance vaccine efficacy.

Microbial Species	Effect to Vaccines	References
*Actinobacteria*	Improved response to BCG, HepB, IPV and OPV vaccination.	Huda and Lépine et al. [71,73]
*Firmicutes bacteria*	Associated with higher humoral responses to ORV in infants and cellular responses to Salmonella Typhi in adults.	Harris et al. [70]
*Bifidobacterium adolescentis*; *Butyricimonas virosa*, *Adlercreutzia equolifaciens*; *Asaccharobacter celatus*	Enhance the efficacy of CoronaVac vaccine.	Siew et al. [63]
*Eubacterium rectale*; *Roseburia faecis*; *Bacteroides*	Enhance the efficacy of BNT162b2 vaccine.	Siew et al. [63]
*Ruminococcus torques*; *Eubacterium ventriosum*; *Streptococcus salivarius*	Associated with high responses to CoronaVac in overweight or obese population.	Siew et al. [63]
*Escherichia coli*	Enhancing the immune response of Influenza Vaccine through the role of TLR5 receptor.	Jason et al. [64]
*Desulfobacterota, Bilophila*	Enhance the immunogenicity of COVID-19 mRNA vaccine (Moderna and BNT162b2) by synthesizing endotoxin with immunostimulatory effect.	Daddi et al. [67]
*Streptococcus bovis*, *Clostridium cluster XI*; *Proteobacteria*	Positively correlated with the enhancement of immune response to ORV vaccine.	Harris et al. [69]
*Clostridiales*	Associated with high abundance of long-term IgG and IgA memory B cells in oral cholera vaccine.	Denise et al. [36]
*Bifidobacterium animalis* ssp. *Lactis*; *Lactobacillus paracasei*	Increased the specific antibody level of influenza vaccine after vaccination.	Rizzardini et al. [74]
*Bifidobacterium infantis*; *Bifidobacterium breve*	Increased titer of anti-poliovirus IgA.	Mullié et al. [73]
*Lactobacillus acidophilus*	Enhanced production of IgA and IgM after OPV vaccination, increased vaccine efficacy against Salmonella Typhimurium strains.	Michael and Lépine et al. [68,72]

BCG, Bacillus Calmette-Guérin; HepB, Hepatitis B; IPV, Inactivated Poliovirus Vaccine; OPV, Oral Poliovirus Vaccine; ORV, Oral Rotavirus Vaccine; IgG, Immunoglobulin G; IgA, Immunoglobulin A; IgM, Immunoglobulin M.

**Table 2 vaccines-12-00868-t002:** Microbial species that may induce vaccine efficacy.

Microbial Species	Effect to Vaccines	References
*Proteobacteria*	Associated with lower responses to BCG, HepB, IPV and OPV vaccination.	Huda et al. [71]
*Bacteroides*	high abundance of Bacteroides is related to the lower humoral response to ORV.	Harris et al. [70]
*Burkholderia*	A large number of TCEM similar to human immunoglobulin V region can reduce the efficacy of the vaccine.	Michael et al. [81]
*Escherichia coli*	Induce the production of non-neutralizing antibodies against HIV gp41 protein.	Williams et al. [82]
*Enterobacterales*	Associated with low abundance of long-term IgG and IgA memory B cells in oral cholera vaccine.	Denise et al. [36]
*Pseudomonadales*	Lower specific T cell response and serum IgG levels were associated with oral OPV vaccination or intramuscular tetanus-hepatitis B vaccine and intradermal BCG vaccine.	Huda et al. [71]
*Streptococcus*	Decrease the efficacy of COVID-19 vaccine (BNT162b2).	Siew et al. [63]
*Bacteroides and Ruminococcus*	Decrease the efficacy of COVID-19 vaccine (CoronaVac).	Siew et al. [63]
*Clostridium; Enterobacter*; *Pseudomonadales*	The serum titers of tetanus toxoid and HepB vaccine specific IgG and IgA decreased.	Huda et al. [85]

BCG, Bacillus Calmette-Guérin; HepB, Hepatitis B; IPV, Inactivated Poliovirus Vaccine; OPV, Oral Poliovirus Vaccine; ORV, Oral Rotavirus Vaccine; TCEM, T cell exposure motifs; IgG, Immunoglobulin G; IgA, Immunoglobulin A.

**Table 3 vaccines-12-00868-t003:** Summary of potential microbial species that may optimize vaccine design.

Microbial Species	Relevant Components and Mechanisms for Optimizing Vaccine Design	References
*Mycobacteria* and *Burkholderia* spp.	TCEMs have been identified and can be leveraged for the development of targeted vaccines.	Robert et al. [81]
*Mycobacterium tuberculosis*	The PE domain of the PE_PGRS33 protein can induce a protective cellular immune response and is a potential vaccine target. It can also be used as the N-terminus of fusion proteins for constructing recombinant vaccines.	Paola et al. [89]
*Enterococci*	A peptide segment of the TMP1 protein from *Enterococci* bacteriophage cross-reacts with tumor-associated antigens, making it a potential candidate for developing anti-cancer vaccines.	Aurélie et al. [99]
*Bifidobacterium breve*	The SVYRYYGL peptide cross-reacts with melanoma antigens and can be utilized for the development of melanoma vaccines.	Catherine et al. [100]
*Bacteroides*	The β-galactosidase in *Bacteroides* contains peptide segments that cross-react with human cardiac myosin heavy chain protein, making it a potential candidate for myocarditis vaccine research.	Cristina et al. [101]
*Akkermansia muciniphila*	Certain peptide segments are associated with multiple sclerosis and can be utilized for the development of potential vaccines.	Sergio et al. [102]
*Enterococcus gallinarum*	Higher titers of anti-Enterococcus gallinarum antibodies are significantly associated with the presence of anti-dsDNA and anti-Sm autoantibodies.	Harini et al. [103]
*Peptostreptococcus anaerobius*	This anaerobic bacterium is enriched in patients with colorectal cancer and may serve as a potential vaccine target.	Long et al. [104]

TCEM, T cell exposure motifs; TMP1, thymidylate synthetase gene.

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
