# Peer review of "Intestinal Microbiota and Its Effect on Vaccine-Induced Immune Amplification and Tolerance"

_vaccines, 2024, doi:10.3390/vaccines12080868_

Round 1

Reviewer 1 Report

Comments and Suggestions for Authors

This review described on Gut Microbiota and its Effect on Vaccine-induced Immune Amplification and Tolerance” is interesting and well written.

1. The information in this review manuscript seems very common and general. For example, in line 273, the expression “as excessive inflammation can lead to immune tolerance” is too general. Therefore, it would be nice if you could provide detailed information about “excessive inflammation”.

 2. The gut is very long and has different roles depending on the region. Additionally, different regions have different microbiomes. Therefore, rather than expressing the intestinal microorganisms, it would be better to express it as the intestinal microorganisms of a specific region.

Author Response

1.We apologize for the lack of detailed mechanistic explanations in our manuscript. We have added explanations of mechanisms or case studies in the following sections:

  • Lines 106-110: Added case studies on the mechanisms by which intestinal microbiota alter the tumor inflammatory environment.
  • Lines 281-285: Added specific mechanisms by which intestinal microbiota influence the immune response to COVID-19 vaccines.
  • Lines 122-135: Added specific cases illustrating the importance of intestinal microbiota homeostasis for immune balance.
  • Lines 446-455: Added cases where multi-layered omics analyses help elucidate the role of intestinal microbiota in disease progression and vaccine development.
  • Lines 448-456: Added specific mechanisms from multi-omics studies on the role of intestinal microbiota in disease progression and vaccine design.

2.Thank you for your suggestion. We have replaced the term "gut microbiota" with "intestinal microbiota" throughout the manuscript.

Reviewer 2 Report

Comments and Suggestions for Authors

Dear authors,

The present paper reunites important information about vaccination.

It underlines the role of gut microbiota in vaccine design and application, showing its input in  vaccine efficacy.

It summarizes also the categories of gut microbiota that may enhance or diminish vaccine efficacy and discusses  vaccine  design and the application of gut microbial characteristics in vaccine development.

Future research should further investigate the use of multi-omics technologies to eluci-19 date the interactive mechanisms between gut microbiota and vaccine-induced immune responses, 20 aiming to optimize and improve vaccine design.

Utilizing immuno-informatics tools, maybe A.I.,  and outer membrane vesicles as novel vaccine carriers may be the vaccination future as well as advance personalized medicine. 

The present review has a proper design and structure.

The figures are very useful, as the tables that show important corelations.

I suggest a third table with the title "Summary of the cited papers according to the reference number" with two columns, one with the summary of the cited reference and the other with the references.

In the summary you should underline the idea of the references cited in the second column.

Citation may be like this: 1sr author et al. {reference number}.

Please also include suggested references.

Author Response

1.Thank you very much for your suggestion. We have revised the citation format in all tables to "1st author et al. {reference number}."

2.Thank you very much for your feedback again. Our manuscript primarily summarizes the categories of intestinal microbiota that may enhance or diminish vaccine efficacy and discusses the application of vaccine design and intestinal microbiota characteristics in vaccine development. In Tables 1 and 2, we present the categories of intestinal microbiota that may enhance or diminish vaccine efficacy and their mechanisms. Based on your suggestion, we have added Table 3 to summarize the microbiota that could potentially optimize vaccine design and their mechanisms.

Reviewer 3 Report

Comments and Suggestions for Authors

While the review is interesting and includes recent information, a thorough review of the English is necessary.Likewise, in the vaccine design section it would be appropriate to include a table as a summary of the latest generation vaccines with their respective components.

Although Figure 2 shows in a general way the design of vaccines that use outer membrane vesicles derived from the microbiota, it would be appropriate to include in a table the different vaccines that have been developed considering the type of bacteria involved, antigens , other components such as polymers/biopolymers and their field of application.

Further delve into multi-omics technologies to elucidate the interactive mechanisms between gut microbiota and vaccine-induced immune responses, in vaccine design.

Review the template since in the footer it appears as Vaccines 2021.

Review more specific points in the attached file.

Comments on the Quality of English Language

Author Response

1. Thank you very much for your suggestion. We have added Table 3 to summarize the microbiota and their associated components that could potentially optimize vaccine design, along with their mechanisms.

2. In the section on multi-omics and gut microbiome research, we have included relevant case studies and mechanisms to elucidate their roles in vaccine design and disease pathogenesis.

3. We deeply apologize for any confusion caused by our previous descriptions. We have revised the language in the PDF as your suggestions.

Round 2

Reviewer 3 Report

Comments and Suggestions for Authors

The only comment is the following:

L.40. Change "is to the HBV vaccine" to "has been that of the HBV vaccine"

Author Response

Thank you very much for your suggestion. We have modified the relevant statements according to your suggestion. Thanks!